# PAX-TS: MODEL-AGNOSTIC MULTI-GRANULAR EXPLANATIONS FOR TIME SERIES FORECASTING VIA LOCALIZED PERTURBATIONS

## ABSTRACT

Time series forecasting has seen considerable improvement during the last years, with transformer models and large language models driving advancements of the state of the art. Modern forecasting models are generally opaque and do not provide explanations for their forecasts, while well-known post-hoc explainability methods like LIME are not suitable for the forecasting context. We propose PAX-TS, a model-agnostic post-hoc algorithm to explain time series forecasting models and their forecasts. Our method is based on localized input perturbations and results in multi-granular explanations. Further, it is able to characterize cross-channel correlations for multivariate time series forecasts. We compare our algorithm with two other state-of-the-art explanation algorithms and present the different explanation types of the method. We found that the explanations of high-performing and low-performing algorithms differ on the same datasets, highlighting that the explanations of PAX-TS effectively capture a model's behavior. Based on time step correlation matrices resulting from the benchmark, we identify 6 classes of patterns that repeatedly occur across different datasets and algorithms. We found that the patterns are indicators of performance, with noticeable differences in forecasting error between the classes. With PAX-TS, time series forecasting models' mechanisms can be illustrated in different levels of detail, and its explanations can be used to answer practical questions on forecasts.

## 1 INTRODUCTION

Time series forecasting models have seen a wave of research over the last years, with subsequent performance improvements, where new generations of models outperform older models. Recently, this has mainly been driven by transformer architectures (Lim et al., 2021; Liu et al., 2023; Naghashi et al., 2025) and foundation models (Das et al., 2024), promising high performance across different forecasting tasks. Forecasting models have also been applied in the industry[1], providing business value for many companies. However, when models are applied to real-time tasks, practitioners and domain experts commonly face challenges in comprehending how and why a model arrived at its forecast. To resolve this issue, explainable artificial intelligence (XAI) (Angelov et al., 2021) is a paradigm that is becoming highly relevant for time series forecasting tasks. However, this problem still remains underexplored, with existing work falling short of providing appropriate and meaningful explanations for AI-based forecasts in time series applications.

When working with end-users, common questions on time series forecasts involve specific time steps or summary statistics, such as: "Why is the forecast of productivity at 15:00 so low?" or "How can we increase the trend of the forecast?". This is even more complex in multivariate forecasting scenarios, where questions are typically focused on cross-channel correlation. Existing explainability methods are not suitable to answer these types of questions and cannot be used in practice to explain time series forecasting models to end-users. To address the lack of explainability methods for forecasting models, new methods must be designed and optimized for the task to produce suitable explanations.

---

[1]https://www.uber.com/blog/forecasting-introduction/

State-of-the-art time series forecasting models (Lin et al., 2023; Liu et al., 2023; Wang et al., 2024; Naghashi et al., 2025) are generally opaque, providing no explanations for their forecasts, making them less intuitive for the end-user. To combat this, existing post-hoc XAI methods like LIME (Ribeiro et al., 2016), which were developed for classification tasks, have been applied to forecasting (Schlegel et al., 2021; Zhang et al., 2023). However, explanation methods designed for classification are generally not suitable for this task because a forecasting model predicts multiple values over time, in contrast to classification models, which output a single class label for each data point.

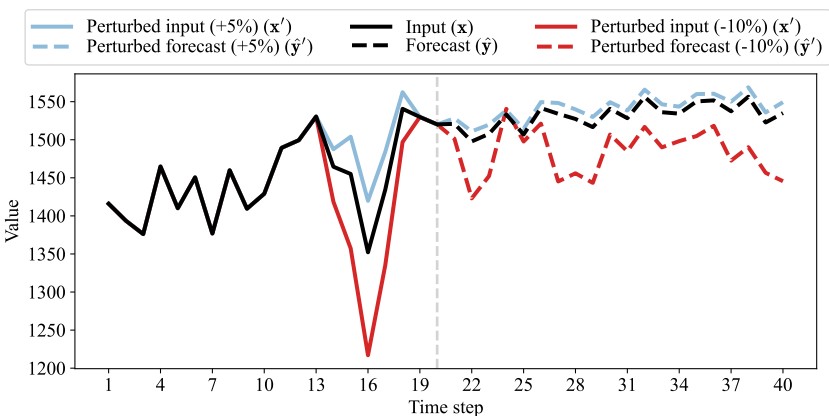

Figure 1: Forecasting with localized perturbations, the basis for `PAX-TS`. The figure shows a perturbation of the global minimum of a sample from *CIF* and the respective forecasts.

In this work, we propose Perturbation Analysis eXplanations for Time Series forecasting (`PAX-TS`), a technique for multi-granular explanations, specific to time series forecasting. `PAX-TS` uses locally perturbed inputs to assess how a forecasting model generates its forecasts, giving insights at different levels of granularity, up to the time-step-level. Fig. 1 shows an example of input perturbations ($\mathbf{x}'$) of the global minimum of an individual sample ($\mathbf{x}$) and the respective forecasts ($\hat{\mathbf{y}}, \hat{\mathbf{y}}'$) for each perturbed input with different scale parameters (+5% and -10%).

Our contributions in this paper are as follows:

1. **Method**: We introduce `PAX-TS`, a post-hoc model-agnostic XAI technique designed for time series forecasting. `PAX-TS` can be used on both univariate and multivariate time series, and its explanations can be generated at different granularities, ranging from high-level cross-channel correlations to low-level time-step-importance explanations.

2. **Evaluation**: We benchmark 7 forecasting models on 10 datasets and compare `PAX-TS` to TS-MULE and ShapTime, then distill six visually distinct temporal-dependency patterns that align with model performance.

3. **Practicality**: We analyze runtime, complexity, and fidelity, showing that `PAX-TS` is suitable for real-time use while accurately characterizing the underlying forecasting model. Code is available at https://anonymous.4open.science/r/pax-ts-6410.

## 2 BACKGROUND

Explainability for time series forecasts can be achieved in two ways: Inherently explainable models are algorithms that are designed to be more comprehensible, providing both the forecast and additional information, which explains the forecast in different ways. Secondly, post-hoc methods can be applied based on the trained model and its forecast to explain how the model has generated the respective forecast. Baur et al. (2024) provide a meta-review on explainability for electric load forecasting, summarising a range of techniques used in this application area. In this section, we describe both inherently explainable forecasting algorithms and post-hoc explanation techniques to give a broad overview of the related work.

**Algorithms providing inherent explanations**. N-BEATS (Oreshkin et al., 2019) is a state-of-the-art univariate time series forecasting algorithm based on multilayer perceptron stacks, which additionally offers a more interpretable architecture. In this architecture, the provided explanations are based on trend and seasonality, which sum up to the model's forecast. This simple yet effective approach allows the end-user to get a more fine-grained understanding of the model's forecast by dividing the forecast into components, which are simpler to comprehend. Lim et al. (2021) introduce temporal fusion transformers, an attention-based forecasting algorithm utilizing past data, static covariates, and known future inputs to make multivariate forecasts. Their algorithm provides values of relative importance to each of the input features. This results in an overall comparison, demonstrating which feature affects the forecast the most. However, this only explains which feature affects a forecast, whether in a positive or negative way, only in a specific segment, or for certain peaks. In a similar way, Pantiskas et al. (2020) use the attention mechanism to generate feature importance scores by time step for their proposed forecasting model. With this method, a specific output time step can be related to the entire input window, showing the importance of each input time step for this specific time step. This forecast explanation method is cumbersome, especially with multivariate data and longer input and output window lengths.

**Post-hoc explanation techniques**. Zhang et al. (2023) propose ShapTime, a method that adapts Shapley Values (Shapley et al., 1953) to characterize the importance of segments in the input sequence. With this method, the importance of each input segment on the mean of the forecast can be assessed, effectively describing how much a given segment affects the forecast overall. While this is useful for some use cases, the granularity of the explanation is too coarse to provide actionable insights. Troncoso-García et al. (2023) propose a method for discovering association rules for univariate time series forecasting. The resulting association rules link single values in the input and the output window with intervals of varying size. This type of explanation is problematic, as many association rules are found, and, in addition to that, their intervals are often large, providing little practical value for the end-user. TS-MULE (Schlegel et al., 2021) is a post-hoc explanation technique for the time series forecasting task, utilizing the perturbation-based approach of LIME (Ribeiro et al., 2016) to make explanations. TS-MULE includes different segmentation techniques to generate explanations, resulting in feature importance scores for segments of the input window. This approach faces problems similar to other feature importance-based techniques like SHAP or attention maps, as the importance of a feature relates to the accuracy of the entire forecast, which is not sufficient to explain time series forecasting models.

## 3 THE PAX-TS ALGORITHM

For the *multivariate time series forecasting* problem, a $d$-dimensional time series subsequence of length $b$, $\mathbf{x} \in \mathbb{R}^{d \times b}$, is to be forecast for a horizon of length $h$, resulting in the forecast $\hat{\mathbf{y}} \in \mathbb{R}^{d \times h}$, aiming to approximate the ground truth $\mathbf{y} \in \mathbb{R}^{d \times h}$. Note that $\mathbf{x}$ and $\mathbf{y}$ are consecutive, and the problem is referred to as *univariate time series forecasting* when $d = 1$. In the literature, $b$ and $h$ vary, where common scenarios range from short horizons $h \leq 48$ to longer horizons of up to $h = 512$. Given an input subsequence $\mathbf{x}$, a trained forecasting model $M_\theta(\cdot)$ generates a forecast $\hat{\mathbf{y}}$, with more accurate forecasting models yielding a closer approximation to the ground truth $\mathbf{y}$, resulting in lower forecasting error. Our proposed approach to explaining time series forecasts, PAX-TS, is perturbation-based and focuses on the relation between input and output subsequences on an index level based on summary statistics and indices of interest.

### 3.1 LOCALIZED PERTURBATION BY INDEX

A localized Gaussian-smoothed perturbation is applied to perturb a positive time series subsequence $\mathbf{x}$ of length $b$ at a given time step $t$, denoted as $x_t$, by a scale parameter $\alpha$, where $\alpha > 0$ indicates a shift in upward direction, and $\alpha < 0$ indicates a shift in downward direction. We introduce this smoothing operation to achieve more plausible perturbations compared to point-based perturbations. Additionally, a width parameter $w \in \mathbb{N}$ affects the perturbation's range, and a softness parameter $s \in \mathbb{R}$ influences the Gaussian fall-off. To apply the perturbation, we first calculate a positional weight $w_p$ for Gaussian-like smoothing, calculated for each time step $i$ of the subsequence. The weight is based on the distance between time step $i$ and $t$, and it is set to 0 for all time steps outside

the fixed window size $w$:

$$w_p \;=\; \begin{cases} \exp\!\left[-\,s\left(\frac{i-t}{w}\right)^2\right], & |i-t| \le w, \\ 0, & |i-t| > w, \end{cases} \qquad \forall\, i \in \{1,\ldots,b\}$$

The amplitude weight is an additional weight in the range $[0,1]$, which downweights values that significantly differ from $\mathbf{x}_t$. It is defined as follows:

$$w_a \;=\; \frac{\min(x_i, x_t)}{\max(x_i, x_t) + \epsilon}, \qquad \forall\, i \in \{1,\ldots,b\}$$

By combining the positional smoothing weight, the amplitude weight, and the scale parameter, the perturbed time series subsequence $\mathbf{x}'$ is obtained as follows:

$$\mathbf{x}' \;=\; \{\, x_i' \,\}_{i=1}^{b}, \qquad x_i' \;=\; x_i \;+\; w_p\, w_a\, \alpha\, x_i, \quad \forall i \in \{1,\ldots,b\}$$

This localized perturbation can be applied at any index $t \in \{1,\ldots,b\}$, which also includes descriptive properties of interest like the maximum or minimum of the time series.

## 3.2 Summary statistic scaling

In this work, we consider the first and second moments of the time series as relevant summary statistics for scaling due to their simplicity and practical relevance. The same procedure can be followed for higher-order moments, however, the equations need to be adapted accordingly. With a scale parameter $\alpha$, the first moment of a time series subsequence $\mathbf{x}$ can be scaled as follows:

$$\mathbf{x}' \;=\; \mathbf{x} + \alpha\mu, \qquad \mu = \frac{1}{b}\sum_{i=1}^{b} x_i$$

Moreover, the second moment can be scaled as follows:

$$\mathbf{x}' \;=\; (\mathbf{x} - \mu) * \sqrt{\alpha + 1} + \mu, \qquad \mu = \frac{1}{b}\sum_{i=1}^{b} x_i$$

## 3.3 Trend-drift adjustment

The drift, representing the overall trend of a time series subsequence $\mathbf{x}$, can be adjusted by deseasonalizing the sequence, fitting a linear polynomial, and adjusting its slope according to a scale parameter $\alpha$. Given a seasonality length $k \in \mathbb{N}$, a discrete convolution operation, denoted by "$*$", results in a deseasonalized moving average of the time series:

$$\mathbf{x}_d \;=\; \frac{1}{k}\, \mathbf{1}_k * \mathbf{x}$$

Using ordinary least squares, a linear polynomial can be fit to the residual $\mathbf{x}_d$, resulting in an intercept $a$ and a slope $m$, representing the trend of the deseasonalized sequence. The adjusted slope can be obtained as follows, while not adjusting the intercept, which provided the highest performance during early experiments:

$$m' \;=\; m + z, \; z \;=\; \begin{cases} \alpha m, & m \ge 0 \\ -\alpha m, & m < 0 \end{cases}$$

Finally, the trend-adjusted subsequence $\mathbf{x}'$ is given by:

$$\mathbf{x}' \;=\; \{\, x_i' \,\}_{i=1}^{b}, \qquad x_i' \;=\; x_i \;-\; (m - m')(i-1), \quad \forall\, i \in \{1,\ldots,b\}.$$

## 3.4 Structured perturbation analysis with PAX-TS

Given a set of scale parameters $\mathcal{A}$, and following the equations of the preceding sections, various characteristics of a time series subsequence $\mathbf{x}$ can be perturbed. In combination with a trained

forecasting model $M_\theta(\cdot)$, forecasts $\hat{\mathbf{y}}'$ can be made for the perturbed input $\mathbf{x}'$. Next, a property of interest function $\pi(\cdot)$ is used, which extracts a human-comprehensible property (such as trend, maximum, or the value at a given index) of a time series. These functions are typically simple, such as $\max(\hat{\mathbf{y}})$ or retrieving the value $\hat{y}_t$, and we do not describe them in further detail. Given the distance $\Delta$ between the perturbed input and the original input, the change ratio $r_{\pi\alpha}$ between a property of interest of the original prediction $\hat{\mathbf{y}}$ and the perturbed prediction $\hat{\mathbf{y}}'$ can be calculated. Using this change ratio, PAX-TS can quantify the effect of applying the perturbation $p(\cdot)$ on the input with relation to a given property of interest $\pi(\cdot)$ of the forecast. When averaging over all scale parameters in $\mathcal{A}$, the mean change ratio $\bar{r}_\pi$ can be calculated, while taking the sign of $\alpha$ into account, as negative scale parameters produce inverse change ratios. This procedure can be applied both to an individual forecasting sample and to an entire dataset by substituting $\mathbf{x}$. Applying PAX-TS for multiple properties of interest on both the input-level and the forecast-level results in a matrix of change ratios, which can explain the behavior of the forecasting model $M_\theta(\cdot)$ in a detailed manner.

## 3.5 Complexity Analysis

To calculate change ratios with PAX-TS, $|\mathcal{A}| + 1$ inference operations need to be performed, which is the most computationally complex part of the method. Additional necessary operations are perturbation, distance calculation, and property of interest extraction. However, as these operations all scale with $O(b)$ and are, therefore, less computationally intensive than the inference operations, they do not contribute to the overall complexity of applying our method. This shows that our method scales linearly with the number of scale parameters $|\mathcal{A}|$, which can be increased to improve the robustness of the method. In practice, $|\mathcal{A}|$ can be set to a low number, e.g. $\leq 4$. As suggested in early experiments, change ratios tend to be constant for different scale parameters when normalized by the distance, so that

$$r_{\pi\alpha_1} - r_{\pi\alpha_2} \leq \epsilon, \qquad \forall (\alpha_1, \alpha_2) \in \mathcal{A},$$
$$\text{where } sgn(\alpha_1) = sgn(\alpha_2).$$

Further, given two inverse scale parameters, $\alpha_1$, and $\alpha_2$, where $\alpha_1 = -\alpha_2$, we found that

$$r_{\pi\alpha_1} + r_{\pi\alpha_2} \leq \epsilon.$$

This finding demonstrates that PAX-TS can be applied to large datasets, as its computational complexity scales linearly with the inference time of the underlying forecasting model, making it suitable for real-time use.

## 3.6 Multivariate cross-channel correlation analysis

PAX-TS enables the analysis of multivariate time series data to assess how a forecasting model takes cross-channel correlations into consideration. To do this, each time series channel is locally perturbed at each input time step. This allows the computation of the change ratio $r_{\pi ct\alpha}$, with respect to the output time step $\pi$, channel $c$, input time step $t$, and scale parameter $\alpha$. When averaging across all input and output time steps, we can calculate $\bar{R}_c$, which is the average change ratio of one channel with respect to all other channels of the time series. Finally, when aggregating the results for all channels, this results in $\bar{R}$, a $d \times d$ matrix of cross-channel correlations. We chose to calculate cross-channel correlations by perturbing all time steps of the input, so that $t \in \{1, \ldots, b\}$, to more accurately capture the forecaster's behavior and account for potential seasonal effects. For datasets with a large number of channels, the computational complexity of multivariate cross-channel correlation can be reduced by setting the number of scale parameters to a low value (e.g., $\alpha = 2$), and timesteps in the input can be changed in a sparsified manner (e.g. $t \in \{1, 3, 5, \ldots, n\}$). However, in the datasets we analyzed in this paper, a reduction was not necessary, as the algorithm generally finished in less than 1 minute per dataset.

# 4 Experiments

## 4.1 Experimental Setup

We test our proposed method with seven forecasting algorithms (see Table 2 in the appendix) on ten datasets, including both univariate and multivariate datasets (see Table 3 in the appendix), both to

demonstrate that the method is model-agnostic and to investigate and explain different types of algorithms for different types of forecasting scenarios. We set the forecasting horizon length to $h = 20$ and input length to $b = 20$ in order to demonstrate a short-term forecasting scenario, where explainability can be highlighted more easily. However, our approach is applicable irrespective of the forecasting scenario and the values of these parameters. We use a temporally ordered, non-overlapping train-validation-test split of 7-1-2, and scale parameters $\mathcal{A} = \{-0.1, -0.05, -0.01, 0.01, 0.05, 0.1\}$ to provide more robust results. We set $w = 2$, $s = 1$, and configure $k$ according to the dataset. We consider three error metrics: The Mean Absolute Error (MAE), and the Mean Squared Error (MSE), which are standard in multivariate forecasting evaluation as well as the Overall Weighted Average (OWA) (Makridakis et al., 2020), which is commonly employed for short-term forecasting evaluation. All experiments and results are publicly available on our anonymous git repository[2], where we share additional insights on `PAX-TS` and its explanations. We conducted a benchmark on the forecasting performance of all models on all datasets, which is presented in Table 5 in the appendix to facilitate readability. The table additionally shows temporal dependency patterns, which we will present in Sec. 4.3.

## 4.2 HIGH GRANULARITY: TIME STEP IMPORTANCE

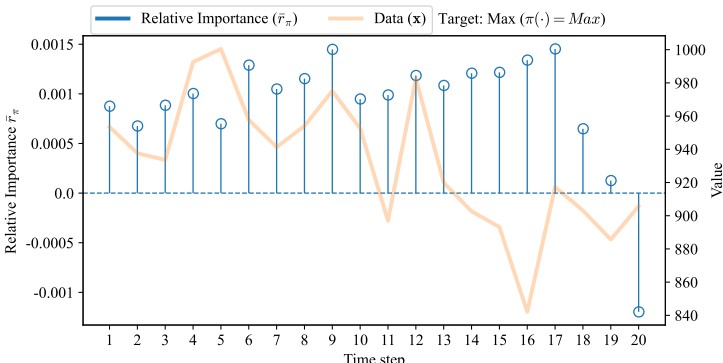

Figure 2: Time step importance of MultiPatchFormer on a sample of the *CIF* dataset with relation to the maximum of the forecast.

`PAX-TS` can be applied on a time-step-level at the highest granularity to highlight the importance of every time step of the input window with relation to a given property of interest of the forecast. For example, Fig. 2 shows an explanation of MultiPatchFormer's (see Naghashi et al. (2025)) forecast on a sample of the *CIF* dataset. Specifically, `PAX-TS` calculates the relative importance $\bar{r}_\pi$ of each input time step for the maximum of the forecast, where $\pi(\cdot) = max$. In the shown example, the first 19 time steps of the input have a positive correlation ($\bar{r}_\pi > 0$) with the maximum, while the last time step has a negative correlation ($\bar{r}_\pi < 0$). With this type of explanation, the end-user can understand which time steps of the data are important, and how the model's predicted maximum changes for different inputs. Setting $\pi(\cdot) = Max$ is an example - when using `PAX-TS` in practice, other properties of interest can be analyzed instead.

## 4.3 MEDIUM GRANULARITY: TEMPORAL DEPENDENCIES

Temporal dependencies at medium granularity are analyzed by using a set of properties of interest $\mathcal{P} = \{\pi_i\}_{i=1}^h$, where $\pi_i = \hat{y}_i'$ and applying `PAX-TS` across all input indices $i \in \{1, \ldots, b\}$, where $p(\cdot)$ perturbs $\mathbf{x}$ at each $i$. The resulting relative-importance matrix visualizes dependencies between input and forecast time steps (Fig. 3). On *Rain*, MultiPatchFormer reaches the best accuracy, whereas SegRNN (Lin et al. (2023)) performs considerably worse. The heatmaps reflect this: SegRNN relies almost exclusively on the last input step, assigning little weight to earlier steps, while MultiPatchFormer exhibits clear seasonal diagonals across the matrix. Because *Rain* is monthly data, the seasonality is $k = 12$, shown by strong positive correlations between, for example, input index 16 and forecast index 20. These patterns characterize model behavior, dataset properties,

---

[2]https://anonymous.4open.science/r/pax-ts-6410

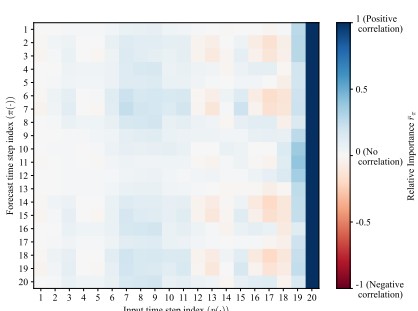 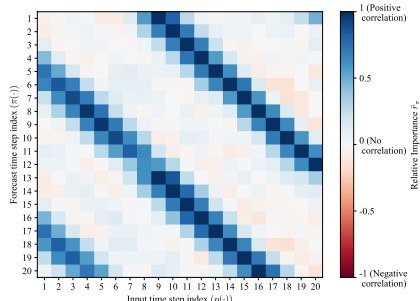

(a) *SegRNN*: Last-time step   (b) *MultiPatchFormer*: Diagonals

Figure 3: Time step correlation on the *Rain* dataset, where MultiPatchFormer achieves high accuracy, while SegRNN performs poorly. The different temporal dependency patterns show that MultiPatchFormer is able to capture dependencies in the data.

| Pattern | Properties | N | Error $e_{norm}$ |
|---|---|---|---|
| Diagonals | Clear diagonal pattern across all timesteps of the input. | 12 | 0.4239 |
| Diagonals (End) | Clear diagonal pattern, repeating over the last seasonal period. | 6 | 0.4645 |
| Last-Timestep | Vertical line on the last timestep, influencing the entire forecast. | 24 | 0.9592 |
| Bipolar Regions | Bipolar positive correlation and negative correlation areas. | 5 | 0.2919 |
| Fully Correlated | Fully positive or negative pattern across all input and output timesteps. | 8 | 0.9690 |
| Other | Other patterns, typically random and noisy. | 5 | 0.7149 |

Table 1: Overview of temporal dependency patterns and their associated error, averaged for all algorithms across all datasets. **N** represents the number of occurrences.

and forecasting quality. Across all benchmarked algorithms and datasets, six temporal-dependency classes were derived (Table 1), which are shown per-dataset in Table 5. Finally, the average error across datasets was computed as a normalized mean of MAE and MSE relative to the naïve forecast as follows:

$$e_{norm} = \frac{1}{2}\left(\frac{MAE_{model}}{MAE_{naive}} + \frac{MSE_{model}}{MSE_{naive}}\right)$$

Diagonals (1) (see Fig. 3b) show a clear diagonal pattern of input-forecast correlation, typically indicating seasonal behavior, and they are a sign of high performance. A similar pattern with slightly lower performance are repeated diagonals in the end (2) of the input window over the last seasonal period. This pattern shows that the model focuses on the last known period of the data (e.g. last 7 days for daily records), and pays little attention to previous input time steps. In some cases, models pay particular attention to the last time step (3), ignoring any other time steps of the input, as SegRNN in Fig. 3a. This is generally an indicator of low performance, with $e_{norm} = 0.9592$, which is close to the performance of a naïve forecast. Bipolar regions (4) are strong regional correlations between input and output, where some regions have positive and other regions have negative correlation. This pattern achieves the highest performance overall across our benchmark and is an indicator of accurate forecasts. In some cases, forecasting models show a fully positive or fully negative correlation (5), so that the time step correlation matrix is almost entirely positive or negative. This pattern is a strong indicator of low forecasting performance, with the highest overall error. Lastly, we categorized some patterns as 'Other' (6), as they were either not fitting any category or a hybrid between categories, achieving mixed performance. For further details on correlation matrix patterns and examples with different datasets, we refer the reader to our repository[3].

## 4.4 Low granularity: Multivariate cross-channel correlations

At the lowest granularity, `PAX-TS`, can assess and visualize cross-channel correlations in multivariate time series forecasting scenarios. This can be done by only perturbing a specific channel of the input and calculating the mean change ratio for all channels of the output separately. Fig. 4 shows the cross-channel correlations, as observed for iTransformer on the *ETTh1* dataset, which consists of

---

[3]https://anonymous.4open.science/r/pax-ts-6410

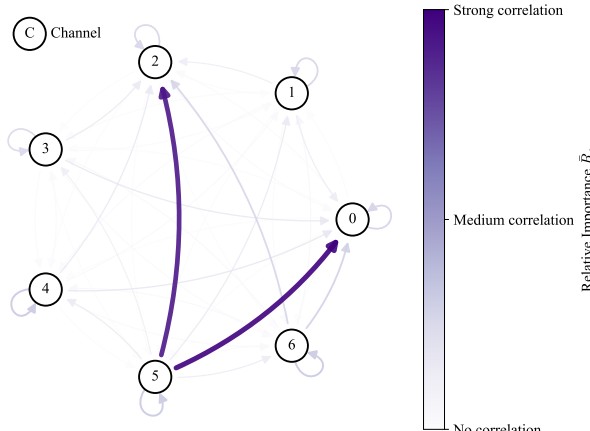

Figure 4: Cross-channel correlations of iTransformer on the *ETTh1* dataset, showing strong relations from channel 5 to channel 0, and from channel 5 to channel 2.

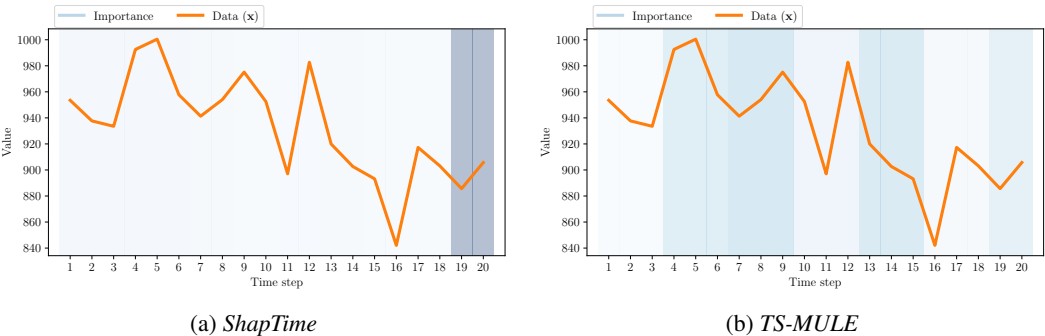

(a) *ShapTime*  (b) *TS-MULE*

Figure 5: High-level input importance explanations of state-of-the art XAI methods for forecasting.

seven channels. We selected this example, as *ETTh1* is the multivariate dataset with the lowest OWA error, where all models perform relatively well. In the visualized explanation, stronger correlations are highlighted with higher line width and more intense color. The model considers two strong cross-channel correlations: Channel 5 affects both channels 0 and 2. These correlations are even stronger than the autocorrelation of channels 0 and 2. Further, we found that on the benchmarked datasets, iTransformer and MultiPatchFormer most frequently utilize cross-channel correlations.

### 4.5 COMPARISON TO SHAPTIME AND TS-MULE

We compare the explanations generated by PAX-TS with two explainability algorithms for forecasting: TS-MULE (Schlegel et al., 2021) and ShapTime (Zhang et al., 2023). Fig. 5 shows explanations as generated by the two methods, following the visualization style of the original papers, for the same sample of the *CIF* dataset previously shown in Fig. 2, where we highlighted high-granularity time step importance in relation to the maximum of the forecast. Both ShapTime and TS-MULE are not able to capture importance for specific properties of interest, such as the maximum of the forecast, while PAX-TS can derive importance scores for different properties of interest, including extrema, summary statistics, and values at a given index. Further, TS-MULE's results are always based on segment importance, rather than timestep importance, having a lower granularity than PAX-TS. The granularity of ShapTime can be configured with the number of segments, up to the time step level. However, as ShapTime evaluates all possible distinct subsets, this results in a time complexity of $O(2^b)$ inference operations, compared to the $O(|\mathcal{A}|b)$ inference operations of PAX-TS to generate importance scores at the time-step-level. Further, given a trained model, both ShapTime and PAX-TS are deterministic, giving them high stability, while TS-MULE produces random results with high variance, leading to high instability. Since ShapTime observes changes in the mean of the forecast, the resulting importance scores are similar to PAX-TS, when setting $\pi(\cdot) = Mean$. Both

ShapTime and TS-MULE are not suitable for application to multivariate data, whereas `PAX-TS` can be used to analyze cross-channel correlations, which produces a correlation graph, as shown in Fig. 4. All explanations produced by `PAX-TS` are directly based on input-output mappings of the forecasting algorithm, giving the explanations a high degree of fidelity, which we empirically prove in the appendix. Both TS-MULE and ShapTime operate on a local scope, giving explanations for an individual forecasting sample. `PAX-TS`, on the other hand, can produce both local and global explanations, increasing its scope of applicability and representativeness.

## 5 DISCUSSION

`PAX-TS` can generate explanations for time series forecasting models at different granularities, which are suitable for different use cases, depending on the aim of the end-user. In Sec. 1, we introduced typical questions from end-users, which we relate to `PAX-TS`' explanations here.

*"Why is the forecast of productivity at 15:00 so low?"*

Answers to questions about specific timestamps can be supported with time-step-level explanations, such as the stemplot in Fig. 2 or the time step correlation matrix shown in Fig. 3. With these explanations, the end-user can clearly pinpoint which time steps of the input are related to the time step of interest and at which magnitude.

*"How can we increase the trend of the forecast?"*

Questions regarding properties of interest, including summary statistics, trend, or extrema, can be addressed in a similar way. Time step importance explanations (Fig. 2) or correlations of different properties of interest (Fig. 6) can provide the end-user with valuable insights to address this.

*"Is there a correlation between productivity and indoor temperature?"*

Lastly, `PAX-TS` can answer questions on cross-channel correlations of multivariate data with a graph, as shown with the example in Fig. 4. For more specific queries between channels, time-step-level index correlation can be used to illustrate the relation between the channels on a higher granularity. To apply `PAX-TS`, a top-down approach with increasingly higher explanation granularity can be employed: For multivariate data, first, a cross-channel correlation graph can be used to unveil correlations between different channels of the data. With this information, a heatmap of index correlations can be visualized to illustrate the correlation between two channels. When the end-user investigates a specific time step, extremum, or summary statistic, a stemplot can be generated to assess the time step importance of the specified sample for the given property of interest.

`PAX-TS` can be applied in a diverse range of application scenarios: In the agriculture domain, it can be used to explain soil moisture forecasts (Deforce et al., 2024), where farmers are aiming to optimize their irrigation system. With our method, the system can be tuned to, for example, increase the minimum moisture level. In the context of a smart building digital twin (Kreuzer et al., 2024), the method can provide detailed explanations for $CO_2$ concentration forecasts to, for example, a facility manager, highlighting how different rooms in the building contribute to the overall $CO_2$ levels. The work on hospital occupancy forecasts by Avinash et al. (2025) could be extended with `PAX-TS` to make use of state-of-the-art forecasters while providing post-hoc explanations for managers.

## 6 CONCLUSION

The explanations provided by `PAX-TS` are both faithful to the forecasting model's outputs and suitable to address practical questions on time series forecasts from end-users. In comparison with ShapTime and TS-MULE, two state-of-the-art XAI methods, `PAX-TS` provides both higher and lower granularity explanations. Further, the method is applicable to multivariate data, where it can highlight cross-channel correlations as well as temporal dependency patterns. In our experiments, we classified the identified temporal dependency patterns into six distinct classes. Across all datasets and algorithms, we found that diagonal and bipolar regional dependency patterns tend to perform best. We believe that future work can apply `PAX-TS` to provide explainability on other datasets and algorithms, which will contribute to our understanding of temporal dependency patterns.

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

| Algorithm | Ref. | Approach |
|---|---|---|
| Naïve | - | Naïve |
| DLinear | Zeng et al. (2023) | MLP |
| MultiPatchFormer | Naghashi et al. (2025) | Transformer |
| SegRNN | Lin et al. (2023) | RNN |
| TimeMixer | Wang et al. (2024) | MLP |
| iTransformer | Liu et al. (2023) | Transformer |
| TimesFM | Das et al. (2024) | LLM |

Table 2: Algorithms evaluated in this study.

Udo Schlegel, Duy Lam Vo, Daniel A Keim, and Daniel Seebacher. Ts-mule: Local interpretable model-agnostic explanations for time series forecast models. In *Joint European conference on machine learning and knowledge discovery in databases*, pp. 5–14. Springer, 2021.

Lloyd S Shapley et al. A value for n-person games. 1953.

Martin Štěpnička and Michal Burda. Computational intelligence in forecasting-the results of the time series forecasting competition. In *2015 IEEE International Conference on Fuzzy Systems (FUZZ-IEEE)*, pp. 1–8. IEEE, 2015.

AR Troncoso-García, María Martínez-Ballesteros, Francisco Martínez-Álvarez, and Alicia Troncoso. A new approach based on association rules to add explainability to time series forecasting models. *Information Fusion*, 94:169–180, 2023.

Shiyu Wang, Haixu Wu, Xiaoming Shi, Tengge Hu, Huakun Luo, Lintao Ma, James Y Zhang, and Jun Zhou. Timemixer: Decomposable multiscale mixing for time series forecasting. *arXiv preprint arXiv:2405.14616*, 2024.

Ailing Zeng, Muxi Chen, Lei Zhang, and Qiang Xu. Are transformers effective for time series forecasting? In *Proceedings of the AAAI conference on artificial intelligence*, volume 37, pp. 11121–11128, 2023.

Yuyi Zhang, Qiushi Sun, Dongfang Qi, Jing Liu, Ruimin Ma, and Ovanes Petrosian. Shaptime: A general xai approach for explainable time series forecasting. In *Proceedings of SAI Intelligent Systems Conference*, pp. 659–673. Springer, 2023.

Haoyi Zhou, Shanghang Zhang, Jieqi Peng, Shuai Zhang, Jianxin Li, Hui Xiong, and Wancai Zhang. Informer: Beyond efficient transformer for long sequence time-series forecasting. In *Proceedings of the AAAI conference on artificial intelligence*, volume 35, pp. 11106–11115, 2021.

## A    EXPERIMENTAL DETAILS

All experiments in the paper were conducted on a single Nvidia A100 GPU, and results were averaged over three random states. All inputs are scaled to have zero mean and unit variance. Table 2 shows an overview of the forecasting algorithms used in this study, which were used as a basis for `PAX-TS` as a post-hoc explainability technique.

Table 3 lists the time series datasets used for the experiments in Sec. 4 as well as their domain, frequency, and the number of variates. In total, 4 multivariate and 6 univariate datasets are used in the experiments.

## B    LOCAL SENSITIVITY ANALYSIS AND FIDELITY

We evaluate the local sensitivity of `PAX-TS` on the *CIF* dataset with DLinear as the underlying forecasting method by comparing the algorithm's averaged change ratios, $\bar{r}_\pi$, to the model's own differential behavior. Concretely, we compute Spearman's $\rho$, cosine similarity, and Kendall's $\tau$ between $\bar{r}_\pi$ and the output Jacobian of DLinear for each horizon timestep of the forecast:

| Dataset | Ref. | Domain | Frequency | Variates |
|---|---|---|---|---|
| M4 Hourly | Makridakis et al. (2020) | Mixed | Hourly | 1 |
| Weather | (a) | Weather | Daily | 1 |
| Transactions | Cook et al. (2021) | Sales | Daily | 1 |
| CIF | Štěpnička & Burda (2015) | Finance | Monthly | 1 |
| Rain | (b) | Weather | Monthly | 1 |
| M4 Yearly | Makridakis et al. (2020) | Mixed | Yearly | 1 |
| Covid | (c) | Healthcare | Weekly | 6 |
| ETTh1 | Zhou et al. (2021) | Electricity | Hourly | 7 |
| Exchange Rate | Lai et al. (2018) | Finance | Daily | 7 |
| Illness | (d) | Healthcare | Weekly | 7 |

Table 3: Datasets used in this study.

(a) https://www.bgc-jena.mpg.de/wetter/,
(b) https://www.kaggle.com/datasets/macaronimutton/mumbai-rainfall-data,
(c) https://www.kaggle.com/datasets/mexwell/sars-cov-2-germany,
(d) https://gis.cdc.gov/grasp/fluview/fluportaldashboard.html

| Setup | Spearman $\rho$ | Cosine similarity | Kendall $\tau$ |
|---|---|---|---|
| Random | -4.19e-4 | -8.90e-4 | -3.23e-4 |
| PAX-TS | 0.778 | 0.872 | 0.575 |
| PAX-TS w/o Gaussian smoothing | **0.986** | **0.993** | **0.926** |

Table 4: Local sensitivity of PAX-TS, compared to random results. We measure Spearman's rank correlation, cosine similarity, and Kendall's tau between the change ratios produced by PAX-TS and the Jacobian of the output of DLinear on the *CIF* dataset, averaged across all samples.

Let a forecasting model be $M_\theta : \mathbb{R}^b \to \mathbb{R}^h$, mapping an input window $\mathbf{x} \in \mathbb{R}^b$ to an $h$-step prediction $\hat{\mathbf{y}} = M_\theta(\mathbf{x}) \in \mathbb{R}^h$. PAX-TS returns a timestep-horizon importance matrix $\bar{R} \in \mathbb{R}^{b \times h}$ with entries $\bar{r}_\pi$ representing averaged change ratios, as described in Sec. 3. We quantify the model's own differential dependence via the (per-sample) Jacobian

$$J[i,j] = \frac{\partial \hat{y}_j}{\partial x_i} \quad \text{for} \quad i = 1, \ldots, b, \ j = 1, \ldots, h,$$

computed by automatic differentiation. For each horizon $j$, define $\mathbf{r}_j = \big(\bar{r}_\pi[1,j], \ldots, \bar{r}_\pi[b,j]\big)^\top$ and $\mathbf{j}_j = \big(J[1,j], \ldots, J[b,j]\big)^\top$. Based on this, we report rank- and angle-based agreement between $\mathbf{r}_j$ and $\mathbf{j}_j$:

$$\text{Spearman's } \rho_j = \rho\big(\text{rank}(\mathbf{r}_j), \text{rank}(\mathbf{j}_j)\big),$$

$$\text{Cosine}_j = \frac{\mathbf{r}_j^\top \mathbf{j}_j}{\|\mathbf{r}_j\|_2 \|\mathbf{j}_j\|_2},$$

$$\text{Kendall's } \tau_j = \tau\big(\mathbf{r}_j, \mathbf{j}_j\big).$$

We aggregate across horizons and across a dataset $\{\mathbf{x}^{(n)}\}_{n=1}^N$ by simple averaging:

$$\bar{\rho} = \frac{1}{Nh} \sum_{n=1}^N \sum_{j=1}^h \rho_j^{(n)}, \quad \overline{\text{Cos}} = \frac{1}{Nh} \sum_{n=1}^N \sum_{j=1}^h \text{Cosine}_j^{(n)}, \quad \bar{\tau} = \frac{1}{Nh} \sum_{n=1}^N \sum_{j=1}^h \tau_j^{(n)}.$$

The resulting coefficients compare PAX-TS' perturbation-based importance $\bar{r}_\pi$ directly to the model's local derivatives with respect to each input timestep. High $\rho$, cosine, and $\tau$ indicate that the timesteps our method marks as influential are exactly those to which the model is most sensitive. This is the appropriate notion of fidelity here, since PAX-TS explains *how the model behaves*, not which timesteps are optimal for forecast accuracy (as in LIME/SHAP). When $p(\mathbf{x}, \alpha, \phi)$ spreads the perturbation to neighboring timesteps via Gaussian smoothing, $\bar{r}_\pi$ incorporates additional context

---

**Algorithm 1** Perturbation analysis of a forecasting sample.

---

1: **Input:** input subsequence $\mathbf{x}$, perturbation function $p(\cdot)$, scale parameters $\mathcal{A}$, perturbation parameters $\phi$, trained forecaster $M_\theta(\cdot)$, property of interest functions $\mathcal{P}$.
2: $\hat{\mathbf{y}} \leftarrow M_\theta(\mathbf{x})$
3: **for all** $\alpha \in \mathcal{A}$ **do**
4:    $\mathbf{x}' \leftarrow p(\mathbf{x}, \alpha, \phi)$
5:    $\Delta \leftarrow dist(\mathbf{x}, \mathbf{x}')$
6:    $\hat{\mathbf{y}}' \leftarrow M_\theta(\mathbf{x}')$
7:    $r_{\pi\alpha} \leftarrow \frac{\pi(\hat{\mathbf{y}}') - \pi(\hat{\mathbf{y}})}{\Delta}, \qquad \forall \pi(\cdot) \in \mathcal{P}$
8: **end for**
9: $\bar{r}_\pi \leftarrow \frac{1}{|\mathcal{A}|} \sum_{\alpha \in \mathcal{A}} \mathrm{sgn}(\alpha)\, r_{\pi\alpha}$
10: **return** $\bar{r}_\pi$

---

**Algorithm 2** Cross-channel correlation analysis.

---

1: **Input:** input subsequence $\mathbf{x}$ ($d > 1$), scale parameters $\mathcal{A}$, perturbation parameters per channel and time step $\phi_{c,t}$, forecasting model $M_\theta(\cdot)$.
2: $\mathcal{P} \leftarrow \{\pi_n : \hat{\mathbf{y}}' \mapsto \hat{y}'_n \mid n = 1, \ldots, h\}$
3: $\hat{\mathbf{y}} \leftarrow M_\theta(\mathbf{x})$
4: **for** $c = 1$ **to** $d$ **do**
5:    **for** $t = 1$ **to** $b$ **do**
6:      **for all** $\alpha \in \mathcal{A}$ **do**
7:        $\mathbf{x}' \leftarrow p(\mathbf{x}, \alpha, \phi_{c,t})$
8:        $\Delta \leftarrow dist(\mathbf{x}, \mathbf{x}')$
9:        $\hat{\mathbf{y}}' \leftarrow M_\theta(\mathbf{x}')$
10:        $r_{\pi ct\alpha} \leftarrow \dfrac{\pi(\hat{\mathbf{y}}') - \pi(\hat{\mathbf{y}})}{\Delta}, \quad \forall \pi \in \mathcal{P}$
11:      **end for**
12:    **end for**
13:    $\bar{R}_{c\alpha} \leftarrow \frac{1}{b} \sum_{t=1}^{b} \Big( \frac{1}{h} \sum_{\pi \in \mathcal{P}} |r_{\pi ct\alpha}| \Big), \quad \forall \alpha \in \mathcal{A}$
14:    $\bar{R}_c \leftarrow \frac{1}{|\mathcal{A}|} \sum_{\alpha \in \mathcal{A}} \bar{R}_{c\alpha}$
15: **end for**
16: **return** $\bar{R} = \{\bar{R}_c\}$.

---

noise. Due to this, rank agreement with the sharp, per-timestep Jacobian may drop slightly. Without smoothing, the perturbation is more localized, yielding the near-perfect alignment reported in Table 4. With smoothing (the default in Algorithm 1), correlations remain very high but are modestly affected.

## C  APPLYING PAX-TS

Algorithm 1 shows a step-by-step guide for applying PAX-TS to a forecasting sample, resulting in the change ratios used as explanations. The procedure is the same as described in Sec. 3.4. In this section, we additionally highlight that PAX-TS is deterministic, based on the steps in algorithm 1. Given a deterministic forecasting model such as a trained deep learning model which does not introduce any inherent randomness, making a forecast with $M_\theta(\cdot)$ is assumed to be deterministic. Further, all permutation methods $p(\cdot)$ introduced in Sec. 3 are inherently deterministic, as seen from the provided equations. This leads to deterministic change ratios $r_{\pi\alpha}$ and mean change ratios $\bar{r}_\pi$, making the outputs of our algorithm deterministic. This is a property that elevates the stability of our method, in comparison to other non-deterministic methods like TS-MULE.

Algorithm 2 details the multivariate cross-channel correlation analysis in a stepwise algorithmic procedure, supporting the reproducibility of this work. With this approach, both multivariate cross-channel correlation heatmaps (as seen in Fig. 7) and cross-channel correlation graphs can be produced (see Fig. 4).

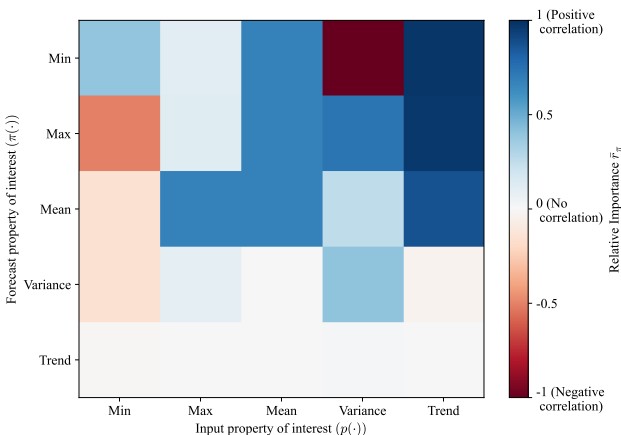

Figure 6: iTransformer on *transactions* showing the relations between minimum, maximum, mean, variance, and trend of the input subsequence and the forecast.

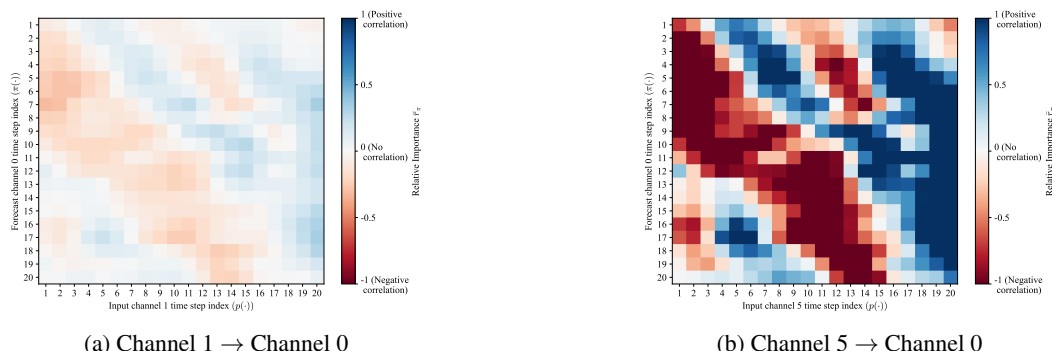

(a) Channel 1 → Channel 0

(b) Channel 5 → Channel 0

Figure 7: Time step correlation matrix of iTransformer on the *ETTh1* dataset. Fig. 7a shows the weak relation from channel 1 to channel 0. Fig. 7b shows the much stronger relation from channel 5 to channel 0.

Fig. 6 shows an example of a correlation matrix between different summary statistics of the *transactions* dataset using iTransformer, as generated by `PAX-TS`. With this type of explanation, `PAX-TS` can answer end-user questions on summary statistics, extrema, and trend of the input and forecast. For example, the figure shows a negative correlation between the input minimum and the output's maximum, mean, and variance. If a user aims to increase the forecast maximum, adjusting the input's mean, variance, and trend appears to be more effective than increasing its maximum value.

## D  MULTIVARIATE CROSS-CHANNEL CORRELATIONS

With `PAX-TS`, the effect of individual pairs of channels can be investigated, and Fig. 7 presents a more fine-grained view of the cross-channel correlations, showing the effect of channels 5 and 1 on channel 0. As already shown in the graph in Fig. 4, channel 5 has a strong effect on channel 0, with both strong negative and strong positive correlations of different time steps. The heatmap visualizes this relation in more detail, highlighting, for example, the importance of the last time step on the entire output window. Channel 1, on the other hand, has a less significant effect on channel 0, with relative importance scores closer to 0 compared to channel 5.

Fig. 8 visualizes an input sample ($\mathbf{x}$) as well as a perturbed sample ($\mathbf{x}'$) of channel 5 of the *ETTh1* dataset. The second row shows the corresponding forecasts of MultiPatchFormer for both samples ($\hat{\mathbf{y}}, \hat{\mathbf{y}}'$) in channel 0. This demonstrates an example where the model does not consider one channel to make a prediction for another one.

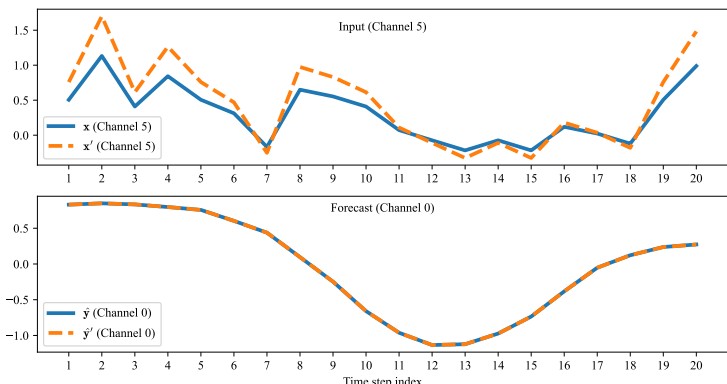

Figure 8: Original and perturbed sample from channel 5 of *ETTh1* (first row), and forecast of channel 0 for both samples (second row), as produced by MultiPatchFormer.

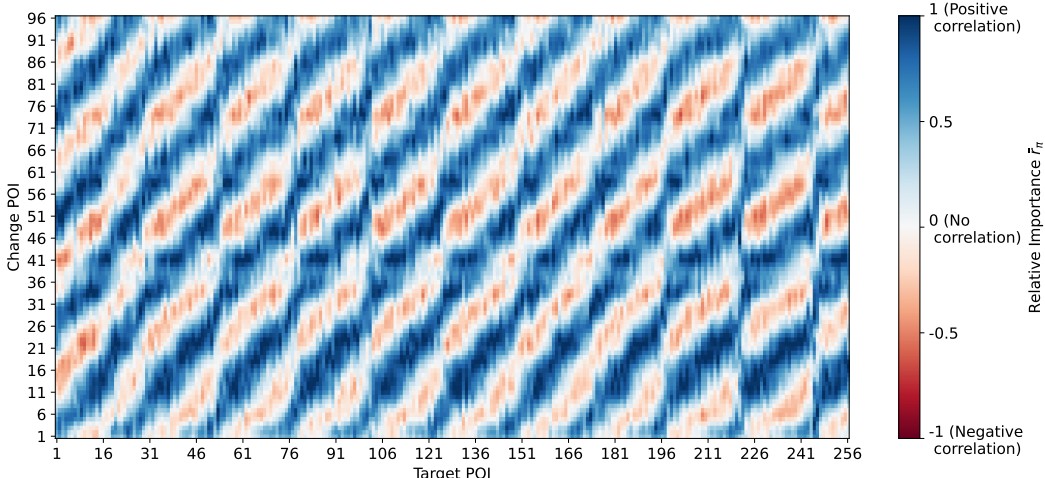

Figure 9: Time step correlation matrix, as produced by PAX-TS, for long-term forecasting, with $b = 96$, and $h = 256$. iTransformer is used as the forecasting model on the dataset *M4 Hourly*.

## E   PAX-TS FOR LONG-TERM FORECASTING

We investigated PAX-TS' capability for explaining forecasting models on long-term forecasts, which is a common evaluation approach in the literature (see e.g. Zhou et al. (2021); Liu et al. (2023); Wang et al. (2024)). Fig. 9 demonstrates an example of a time step correlation matrix for iTransformer on the *M4 Hourly* dataset, produced by PAX-TS. The matrix shows the bipolar regions pattern, across a long-term horizon, reflected by the positive and negative change ratios with clear structure. This illustrates the applicability of our method to time series forecasting setups of any combination of input window length $b$ and forecasting horizon $h$. This further demonstrates that all methods and visualizations presented in this paper are applicable to both short-term and long-term forecasting setups.

## F   FORECASTING ERROR AND TEMPORAL DEPENDENCY PATTERNS

Table 5 gives a detailed overview of all evaluated forecasting models' performance on each dataset. The table further highlights which temporal dependency pattern each model showed when analyzing a time step correlation matrix, such as the ones shown in Fig. 3. The last row of Table 5 shows the mean rank of each model across all datasets as well as the p-value, showing whether the performance is significantly different from the naïve forecasting model based on the Bonferroni-Dunn

test. Notably, TimeMixer, MultiPatchFormer, and iTransformer perform best, with a significant Bonferroni-Dunn test (Dunn, 1961) compared to the Naïve model. On the multivariate datasets *Covid*, *Exchange Rate*, and *Illness*, none of the benchmarked models achieves considerably better performance than the naïve forecaster, highlighting the difficulty of the multivariate forecasting problem. We additionally measured the total inference time across all datasets (Algorithm 1 line (2) + line (6)), where the LLM TimesFM incurred a significantly higher time (1390s) compared to TimeMixer (20.57s), MultiPatchFormer (8.89s), and iTransformer (1.08s).

| Models | | Naïve | DLinear | MultiPatchFormer | SegRNN | TimeMixer | iTransformer | TimesFM |
|---|---|---|---|---|---|---|---|---|
| **M4(H)** | MAE | 0.037 | **0.007** | 0.0078 | 0.012 | 0.0071 | 0.0079 | 0.0084 |
| | MSE | 0.066 | 0.0022 | **0.0018** | 0.0067 | 0.002 | 0.0019 | 0.0044 |
| | OWA | 1 | **0.31** | 2.6 | 1.4 | 0.57 | 3.6 | 0.51 |
| | **Pattern** | - | (D) | (D) | (F) | (D) | (D) | (A) |
| **Weath.** | MAE | 0.39 | **0.36** | 0.37 | 0.36 | 0.36 | 0.37 | 0.39 |
| | MSE | 0.25 | **0.21** | 0.22 | 0.22 | 0.22 | 0.22 | 0.25 |
| | OWA | 1 | **0.94** | 0.95 | 0.95 | 0.95 | 0.95 | 0.99 |
| | **Pattern** | - | (C) | (C) | (C) | (C) | (C) | (E) |
| **Trans.** | MAE | 0.3 | 0.17 | 0.17 | 0.18 | 0.16 | **0.16** | 0.17 |
| | MSE | 0.24 | 0.096 | 0.084 | 0.093 | 0.08 | **0.078** | 0.11 |
| | OWA | 1 | 0.62 | 0.63 | 0.68 | 0.61 | **0.6** | 0.63 |
| | **Pattern** | - | (B) | (B) | (B) | (B) | (B) | (B) |
| **CIF** | MAE | **0.02** | 0.029 | 0.022 | 0.022 | 0.023 | 0.021 | 0.023 |
| | MSE | 0.043 | 0.088 | 0.043 | 0.047 | 0.051 | **0.039** | 0.047 |
| | OWA | **1** | 57 | 36 | 18 | 4.2 | 39 | 15 |
| | **Pattern** | - | (E) | (C) | (C) | (F) | (E) | (E) |
| **Rain** | MAE | 1.1 | 0.41 | 0.37 | 0.8 | 0.42 | **0.37** | 0.64 |
| | MSE | 2.7 | 0.45 | 0.39 | 1.2 | 0.44 | **0.39** | 1.3 |
| | OWA | 1 | 0.45 | **0.39** | 1 | 0.46 | 0.4 | 0.64 |
| | **Pattern** | - | (A) | (A) | (C) | (A) | (A) | (A) |
| **M4(Y)** | MAE | 0.21 | 0.24 | 0.2 | 0.21 | 0.2 | **0.2** | 0.2 |
| | MSE | 0.34 | 0.35 | 0.33 | 0.34 | **0.32** | 0.33 | 0.33 |
| | OWA | 1 | 1.7 | 0.97 | 1.2 | 0.96 | **0.96** | 0.97 |
| | **Pattern** | - | (C) | (C) | (C) | (C) | (C) | (C) |
| **Covid** | MAE | 0.5 | 0.5 | 0.46 | 0.55 | 0.46 | 0.49 | **0.39** |
| | MSE | 0.81 | 0.52 | 0.55 | 0.79 | **0.52** | 0.59 | 0.59 |
| | OWA | 1 | 1.2 | 0.98 | 1.2 | 0.99 | 1 | **0.86** |
| | **Pattern** | - | (E) | (F) | (C) | (E) | (E) | (A) |
| **ETTh1** | MAE | 0.84 | 0.41 | **0.41** | 0.54 | 0.41 | 0.43 | 0.51 |
| | MSE | 1.6 | 0.38 | **0.38** | 0.62 | 0.39 | 0.41 | 0.65 |
| | OWA | 1 | 0.66 | **0.66** | 0.8 | 0.67 | 0.69 | 0.74 |
| | **Pattern** | - | (A) | (A) | (F) | (A) | (A) | (E) |
| **Exch.** | MAE | **0.075** | 0.078 | 0.076 | 0.077 | 0.076 | 0.077 | 0.083 |
| | MSE | **0.013** | 0.013 | 0.013 | 0.013 | 0.013 | 0.013 | 0.015 |
| | OWA | **1** | 1 | 1 | 1 | 1 | 1 | 1.1 |
| | **Pattern** | - | (C) | (C) | (C) | (C) | (C) | (A) |
| **Illness** | MAE | 1.4 | 1.5 | 1.3 | 1.4 | 1.4 | **1.3** | 1.6 |
| | MSE | 5 | 4.6 | 3.8 | 4.5 | 4.1 | **3.6** | 6.5 |
| | OWA | **1** | 1.3 | 1.1 | 1 | 1.1 | 1 | 1.1 |
| | **Pattern** | - | (D) | (C) | (C) | (F) | (C) | (C) |
| | Rank | 5.17±2.32 | 4.27±2.22 | 2.93±1.29 | 4.87±1.45 | **2.87**±1.20 | 3.03±1.80 | 4.87±1.71 |
| | p-Value | - | 0.640 | **0.000** | 1.000 | **0.000** | **0.001** | 1.000 |

Table 5: Errors of the evaluated algorithms on each dataset and corresponding temporal dependency pattern. (A) Diagonals, (B) - Diagonals (End), (C) - Last-Timestep, (D) - Bipolar Regions, (E) - Fully correlated, (F) - Other. The best-performing algorithm for each metric is highlighted in bold. p-Values are calculated using the Bonferroni-Dunn test compared to the Naïve model, with significance ($p < 0.05$) highlighted in bold.

