# OpenReview forum: "PAX-TS: Model-agnostic multi-granular explanations for time series forecasting via localized perturbations"
_ICLR.cc/2026/Conference — ICLR 2026 Conference Withdrawn Submission_

### Official Review · Reviewer_A6yo · 2025-10-21

**Soundness:** 2
**Presentation:** 3
**Contribution:** 2
**Rating:** 4
**Confidence:** 3

**Summary:**

This paper introduces a model-agnostic perturbation-based framework for explaining time series forecasting models at multiple granularities.

**Strengths:**

1. The idea of using localized Gaussian perturbations to derive fine-grained importance maps is interesting.

2. The paper is clear-written.

**Weaknesses:**

1. The paper states that PAX-TS provides multi-granular explanations, but this concept is not clearly defined. It remains unclear whether the granularity refers to temporal resolution, statistical properties, or interpretive abstraction levels.

2. In experiment part, while the paper positions PAX-TS as superior to existing explainers, the experimental evidence is mostly qualitative. Providing quantitative evaluation would strengthen model credibility.

3. The introduction of width 𝑤, softness 𝑠, and scaling 𝛼 parameters is heuristic. Adding a sensitivity analysis to show how these choices affect explainability would be valuable.

4. The categorization of temporal-dependency patterns seems manually defined and lacks quantitative validation.

**Questions:**

Please refer to weaknesses

---

### Official Review · Reviewer_RPSs · 2025-10-27

**Soundness:** 3
**Presentation:** 2
**Contribution:** 2
**Rating:** 2
**Confidence:** 2

**Summary:**

The paper introduces PAX-TS, a simple algorithm to explain time series forecasts.

**Strengths:**

- The explanation methods seem to be simple enough and explainable itself. The methods are more based on traditional statistical properties of time series.

**Weaknesses:**

- Notations and methods are not thoroughly included. Methods are not carefully described.
- As I am not an expert of time series forecasting, I am not confident about the following statement - I think the methods included for comparison are not SOTA. Comparing with other time series explainability papers (which are not limiting to forecasting), for example FIT, Dynamask, WinIT, TimeX, etc., the other methods chosen are ShapTime and TS-MULE. TS-MULE was from 4 years ago and ShapTime is not available for me. The paper also lacks quantitive results for these comparisons.
- Several parts need involvements from human. For example, the "Pattern classification" in Figure 3., or choosing the seasonality constant k.

**Questions:**

As I mentioned in the Weakness section, this paper is very different from the other papers I encountered in the field. I think the style is more suited for a research journal or a thesis, as there is no quantitative comparisons with other methods. The method section also seems to be written in a more academic way, rather than a machine learning way.

Nevertheless, I have several questions and comments

- Section 3.1, 3.2 and 3.3 corresponds to perturbations of the original time series. Are they all being applied? I suppose all these adjustments are not commutative. Is there an order of applications?

- Section 3.1, it is better to say fixing t, $w_{p, i}=...$, $w_{a, i}=...$, if I understand correctly.
- Section 3.4, it is not clear what the interest function $\pi$ represents. For example, if $\pi=max$, what is that being "maximum" with respect to? Is it across all the d-th and h-th dimensions (i.e. all the features and all the time points in the forecasting window?)
- Section 3.4. The change ratio $r_{\pi\alpha}$ is not clear. Is there a formula of this change ratio with respect to the original prediction $\hat{y}$ and the perturbed prediction $\hat{y}'$? Will there be a problem if the whole time series is just shifted vertically (thus the "ratio" would be different)?

- Section 4.2 and Figure 2. This is an example I do not understand. The setting is h=20 and b=20. Thus the 20 time steps are from the input length? Without knowing what the forecasting value, how do we interpret Figure 2? Also, why is the first 19 time steps having a positive correlation with the maximum while the last timesteps having a negative correlation with the maximum?

- Section 4.3, the definition of $\pi_i = \hat{y}_i'$ is not clear. I think normally, when $y$ is a vector, $y_i$ corresponds to its i-th component. But here, I think $\hat{y}_i'$ corresponds to the perturbed output when the input at time $i$ is perturbed?

---

### Official Review · Reviewer_gsqo · 2025-10-29

**Soundness:** 2
**Presentation:** 1
**Contribution:** 2
**Rating:** 2
**Confidence:** 4

**Summary:**

The paper introduces **PAX-TS**, a model-agnostic post-hoc algorithm that explains time series forecasting models through localized input perturbations, providing multi-granular insights. It captures cross-channel correlations in multivariate forecasts and distinguishes the behaviors of high- and low-performing models on the same datasets. By analyzing time-step correlation matrices, the authors identify six recurring explanation patterns across datasets and algorithms.

**Strengths:**

The patterns discovered in the experiments are intriguing and offer valuable insights. It would be interesting to explore how generalizable these patterns are and whether they could inform the design of inherently interpretable time-series forecasting models.

**Weaknesses:**

The provided code link is inaccessible, preventing reproducibility. The formalization section is difficult to follow—Section 3.1 describes localized perturbations without pseudocode or formulas, leaving room for ambiguity. It is also unclear whether Sections 3.2 and 3.3 present original contributions or restate existing concepts, and the inclusion of a running example would greatly improve clarity. Moreover, the experimental objectives are not clearly defined, and the presentation of results is fragmented and lacks coherence, requiring better organization and control.

**Questions:**

Questions can be derived from the above weaknesses.

---

### Official Review · Reviewer_PAYz · 2025-10-30

**Soundness:** 3
**Presentation:** 2
**Contribution:** 2
**Rating:** 4
**Confidence:** 3

**Summary:**

This paper introduces PAX-TS, a model-agnostic explainability framework for time series forecasting based on localized input perturbations. Specifically, it generates multi-granular explanations to help users understand how forecasts are formed. Through experiments on diverse time series forecasting models and multiple datasets, the authors show that PAX-TS provides explanations that outperform existing methods.

**Strengths:**

- The paper addresses a practically important problem in time series forecasting.
- PAX-TS is model-agnostic. The authors tested PAX-TS on various time series forecasting backbones.
- Experiments are extensively conducted using multiple datasets.

**Weaknesses:**

- Some existing approaches for time series interpretation are not discussed or compared:
  - Prototype-based time series (or sequence) interpretability methods
    - Interpretable and steerable sequence learning via prototypes (KDD 2019)
    - Interpreting Convolutional Sequence Model by Learning Local Prototypes with Adaptation Regularization (CIKM 2021)
    - Protgnn: Towards self-explaining graph neural networks (AAAI 2022)
  - LLM-based interpretable methods
    - Explainable Multi-modal Time Series Prediction with LLM-in-the-Loop (NeurIPS 2025)
    - TimeCAP: Learning to Contextualize, Augment, and Predict Time Series Events with Large Language Model Agents (AAAI 2025)
- Figure 1 in the introduction section is not interpretable. Please elaborate on this example.
- The experiments lack a quantitative evaluation of explanation quality.
- While the authors claim that PAX-TS helps answer practical user questions, there is no human validation to support this. Using LLMs can be one potential approach to address this limitation.
- Most experiments focus on short-term forecasting. Recently, more methods focus on long-term forecasting (e.g., output length of 512).

**Questions:**

Please refer to the weaknesses listed above.

---

### Note · Authors · 2025-11-28

**Comment:**

We hereby withdraw our paper from the conference.

**Withdrawal Confirmation:**

I have read and agree with the venue's withdrawal policy on behalf of myself and my co-authors.